# JDAPCOO: Resource Scheduling and Energy Efficiency Optimization in 5G and Satellite Converged Networks for Power Transmission and Distribution Scenarios

**DOI:** 10.3390/s22187085

**Published:** 2022-09-19

**Authors:** Sachula Meng, Sicheng Zhu, Zhihui Wang, Ruibing Zhang, Jinxia Han, Jiayan Liu, Haoran Sun, Peng Qin, Xiongwen Zhao

**Affiliations:** 1Information & Communication Department, China Electric Power Research Institute, Beijing 100192, China; 2School of Electrical and Electronic Engineering, North China Electric Power University, Beijing 102206, China

**Keywords:** 5G and satellite converged network, energy efficiency, device association, power control

## Abstract

Along with the continuous revolution of energy production and energy consumption structures, the information data of smart grids have exploded, and effective solutions are urgently needed to solve the problem of power devices resource scheduling and energy efficiency optimization. In this paper, we propose a fifth generation (5G) and satellite converged network architecture for power transmission and distribution scenarios, where power transmission and distribution devices (PDs) can choose to forward power data to a cloud server data center via ground networks or space-based networks for power grid regulation and control. We propose a Joint Device Association and Power Control Online Optimization (JDAPCOO) algorithm to maximize the long-term system energy efficiency while guaranteeing the minimum transmission rate requirement of PDs. Since the formulated issue is a mixed integer nonconvex optimization problem with high complexity, we decompose the original problem into two subproblems, i.e., device association and power control, which are solved using a genetic algorithm and improved simulated annealing algorithm, respectively. Numerical simulation results show that when the number of PDs is 50, the proposed algorithm can improve the system energy efficiency by 105%, 545.05% and 835.26%, respectively, compared with the equal power allocation algorithm, random power allocation algorithm and random device association algorithm.

## 1. Introduction

With the extensive deployment of “new energy and new business”, the production and operation links covered by smart grids continue to increase. In power transmission and distribution application scenarios such as unmanned aerial vehicle (UAV) transmission line inspection, robot power facility inspection, and emergency communication, etc., the power transmission and distribution devices (PDs) will generate a large amount of data such as monitoring data and video data, which need to be uploaded to the cloud server data center in time for power grid regulation and control [1,2,3]. Facing the trend of explosive growth of power transmission and distribution device information data, the traditional ground network data collection method will not be able to carry massive data transmission; therefore, the existing power communication method needs to be upgraded to monitor more production links and accommodate more power data.

Building a fifth generation (5G) and satellite converged network is a feasible way to solve the above-mentioned challenges. The 5G network is based on a software-defined network [4], network function virtualization [5] and other technologies, which can support on-demand customization, high dynamic expansion, and the automated deployment of network resources. Applying 5G communication technology to power transmission and distribution scenarios can further improve the system performance of wireless private networks and enhance the differentiated and secure bearing capacity of multiple services. Meanwhile, the Internet of Satellite (IoS) has developed rapidly in recent years, which can provide navigation and positioning, precise timing, short message communication and other services for the construction of power transmission and distribution scenarios. IoS usually uses a Ka or Ku frequency band [6] to make the system capacity increase significantly, and it can provide high-speed broadband Internet access services for areas where traditional Internet erection costs are too expensive or ground base stations cannot provide coverage. Furthermore, ref. [7] also studied the performance limits of cognitive-uplink fixed satellite service (FSS) and terrestrial fixed service (FS) operating in the range 27.5–29.5 GHz for the Ka band, which provided useful guidance for system design and performance evaluation. Therefore, the fusion of a 5G network and satellite network applied to power transmission and distribution scenarios can accommodate more power data and meet the bearing needs of power business.

However, the resource scheduling and energy efficiency optimization of a 5G and satellite converged network for power transmission and distribution scenarios still faces some key issues. Firstly, since satellites are far away from ground PDs, the direct uploading of data from ground to satellite may lead to large transmission loss; at present, the research studies on data transmission in 5G and a satellite converged network rarely consider the advantages of UAVs, while the introduction of a UAV in 5G and the satellite converged network of a UAV as a relay for data transmission [8] as well as the data forwarding by the UAV through the designated air-space link can appropriately reduce this transmission loss. Therefore, it is necessary to consider adding a UAV to the 5G and satellite converged network architecture for power transmission and distribution scenarios. Secondly, device association is an essential technology for improving system energy efficiency in 5G and satellite converged heterogeneous networks [9]; therefore, a reasonable device association strategy needs to be designed. Finally, power control techniques play an important role in reducing device energy consumption, which has a significant impact on system performance [10]; further, joint optimization of device association and power control can better improve system performance and user experience.

In summary, this paper investigates the resource scheduling and energy efficiency optimization problem in a 5G and satellite converged network for power transmission and distribution scenarios. We propose a Joint Device Association and Power Control Online Optimization (JDAPCOO) algorithm to maximize the long-term total system energy efficiency while guaranteeing the minimum transmission rate requirement of PDs. The contributions are as follows.

Combined with the actual application scenarios of PDs, this paper proposes a 5G and satellite converged network architecture for power transmission and distribution scenarios, where PDs can choose to forward power data to cloud server data centers via ground networks or space-based networks. Among them, a UAV is introduced as a relay between PDs and satellite to ensure the stability of system transmission.On the premise of ensuring the minimum transmission rate requirements of PDs, we propose an online optimization algorithm of joint device association and power control, including a device association strategy based on a genetic algorithm and device power control scheme based on an improved simulated annealing algorithm. By solving the device association strategy and power control scheme in each time slot, the long-term total system energy efficiency is maximized.We conduct extensive simulations to compare our algorithm with several benchmark algorithms. The results show that our solution has better performance.

The rest of this paper is organized as follows. Section 2 provides an overview of related works. Section 3 presents the system model and problem modeling. The problem solution will be introduced in Section 4. Section 5 presents the numerical results of the simulation. Section 6 is the conclusion of the paper.

## 2. Related Works

At present, some standardization organizations and universities have carried out research on 5G and satellite converged network architecture. In [11], Kapovits et al. studied the feasibility of seamless and efficient integration of ground communication systems with satellite networks. The third Generation Partnership Project (3GPP) standardization document puts forward four preliminary network architecture models for satellite-ground convergence [12], and it discusses the adaptive modifications that need to be made to deploy 5G New Radio (NR) in non-ground networks. Ge et al. presented a multi-access edge computing 5G and satellite converged network architecture supporting enhanced mobile broadband, and they leveraged the proposed architecture to guarantee the quality of experience for streaming media users in [13]. Boero et al. introduced an SDN-based ground-satellite network architecture and estimated the average transmission delay and control delay in [14]. Lin et al. focused on the joint beamforming design and optimization of the reconfigurable intelligent surface (RIS) assisted hybrid satellite-terrestrial relay networks to minimize the total transmit power of the satellite and base station while ensuring the user rate requirements in [15]. An et al. formulated a constrained optimization problem to maximize the instantaneous rate of the terrestrial user while satisfying the interference probability constraint of the satellite user in [16], and they studied the secrecy performance of the cognitive satellite terrestrial network.

Although there have been some research achievements on the 5G and satellite communication convergence at home and abroad, there are few researches on the business and application scenarios of power transmission and distribution scenarios. Moreover, the network architecture of the above literature only includes terrestrial infrastructure and satellites, ignoring the advantages of UAVs. Different from the above work, we also take advantage of the high mobility of UAVs [17,18] and use a UAV as a relay device for ground and satellite communication, which can further improve the quality of data transmission.

In heterogeneous networks, device association is an important issue to improve system performance, and the goal of the device association issue is to determine the communication mode of every ground device in heterogeneous networks [19,20]. Kaleem et al. proposed a user association scheme based on the public security user priority to solve the problem of user association in a multi-layer heterogeneous network, and they realized load balancing and interference management in a long-term evolution system of high volatility public security in [21]. Mlika et al. studied the strategy of base station dormancy and user association under the constraint of guaranteeing the minimum rate of users in [22]. The proposal of this scheme can effectively shut down the base station in the standby state, thereby achieving the purpose of saving the energy consumption of the entire heterogeneous network. In addition, from the perspective of energy saving, power control technology plays an important role in reducing device energy consumption [23,24]. Zhang et al. proposed an energy-efficient power allocation and wireless backhaul bandwidth allocation method under the constraints of specific quality of service (QoS) to maximize the system energy efficiency of downlink heterogeneous networks in [25]. Qiu et al. studied a gradient-based iterative algorithm to find the optimal solution for energy-saving power allocation, thereby maximizing the energy efficiency of the system in [26]. Efrem et al. studied the weighted sum of energy efficiency in heterogeneous networks considering power and rate constraints in [27].

Although relevant studies on device association or power control have been carried out in the above literature, there are few studies on the joint optimization of device association and power control for 5G and satellite converged networks oriented to power transmission and distribution scenarios. Therefore, inspired by the above work, in order to further improve the system performance and user experience, the joint optimization problem between user association and power control in heterogeneous networks also becomes particularly important. Inspired by the above work, under the premise of ensuring the minimum transmission rate requirement of PDs, we propose an online optimization algorithm that combines device association and power control to maximize the long-term total system energy efficiency.

## 3. System Model and Problem Modeling

### 3.1. System Model

The power transmission and distribution scenario is composed of a low earth orbit (LEO) satellite, a UAV, a ground 5G base station, and *I* ground PDs, as shown in Figure 1. Specifically, the PDs i∈{1,…,i,…,I} on the ground run different power services, thereby generating various power data, such as acquisition data, monitoring data, video data, etc., which need to be uploaded to the cloud server data center. In the model, the ground 5G base station is equipped with a cloud server data center, and the PDs can upload the collected data to the ground 5G base station. Due to the fixed locations of the PDs and the 5G base station, the link loss of data transmission for PDs with longer distance is large and consumes too much transmission energy; the actual amount of data transmitted to the base station will be consistently low, which seriously affects the information collection of PDs. In addition, a large number of PDs uploading data to the ground 5G base station will also lead to an excessive transmission link load on the 5G base station. Therefore, in order to reduce the link load and improve the energy efficiency of data transmission, we provide another way of data transmission, i.e., taking advantage of the UAV and the satellite. PDs upload data to the LEO satellite via the UAV. Among them, the UAV operates on a predetermined trajectory as a relay for data forwarding, and the LEO satellite, similar to the ground 5G base station, is equipped with a cloud server. The components participating in the system are summarized as follows.

Power transmission and distribution device (PD): run different power services and generate a variety of power data that need to be uploaded to the cloud server data center.UAV: as a relay for data forwarding of PDs, which is used to forward the power data from the PDs to the cloud server data center of an LEO satellite.5G base station: equipped with a cloud server data center, which is used to receive power data from PDs that need to be uploaded to the cloud server data center.LEO satellite: equipped with a cloud server data center to receive power data forwarded by the UAV that need to be uploaded to the cloud server data center.

In addition, we show in Figure 2 how to communicate the components participating in the system according to Figure 1.

We adopt a time-slotted model to formulate the network operation time, which is evenly divided into *T* time slots. Therefore, the duration of each time slot is τ and the entire operation time can be expressed by {1,…,t,…,T}. Since the UAV moves, the distance $d_{i,u}\left(t\right)$ between the PD and the UAV changes among slots, while the distance $d_{i,b}$ between PD and the ground 5G base station remains constant.

According to the two data transmission ways of PD provided above, we use binary variables $x_{i,u}\left(t\right)$ and $x_{i,b}\left(t\right)$ to represent the association between the PD *i* and the UAV as well as the ground 5G base station, respectively. $x_{i,u}\left(t\right)=1$ means that the PD chooses to upload data to the UAV and then forward it to the LEO satellite in the current time slot, while $x_{i,u}\left(t\right)=0$ means that the PD does not choose to upload the data to the UAV. Similarly, $x_{i,b}\left(t\right)=1$ means the PD in the current time slot chooses to directly forward the data to the ground 5G base station, while $x_{i,b}\left(t\right)=0$ means PD *i* does not choose to forward the data to the ground 5G base station. In particular, to avoid wasting excessive resources by repeatedly forwarding data, each PD can only choose one data transmission way in each time slot, i.e.,
(1)$$x_{i,u}\left(t\right)+x_{i,b}\left(t\right)\leq 1.$$

When the PD chooses to transmit the data via the UAV, it is assumed that the flight height $H_{u}$ of the UAV is the minimum height that satisfies avoiding the actual terrain or buildings. Therefore, there is no need to adjust the flight height of the UAV frequently, and the severe ground attenuation of the UAV during sensing and transmission can be ignored [28]. Meanwhile, considering that the Doppler effect generated by the UAV motion is compensated [29], the channel gain from the PD to the UAV is consistent with the model of free-space path loss [30], which can be expressed as
(2)$$h_{i,u}\left(t\right)=\frac{h_{0}}{{\left[d_{i,u}\left(t\right)\right]}^{2}},$$
where $h_{0}$ denotes the channel gain per unit distance. Furthermore, since the distance between the UAV and the LEO satellite is far away, the horizontal distance can be negligible; then, the channel gain between the UAV and the LEO satellite still conforms to the free-space path loss model and can be represented as
(3)$$h_{u}=\frac{h_{0}}{{\left[H_{g}-H_{u}\right]}^{2}},$$
where $H_{g}$ denotes the altitude of the LEO satellite from the ground. Let $R_{i,u}\left(t\right)$ represent the data rate of the current time slot PD uploaded to the LEO satellite, which can be expressed as
(4)$$R_{i,u}\left(t\right)=W\cdot \log_{2}\left(1+\frac{\gamma_{i,u}\left(t\right)\cdot \gamma_{u}}{1+\gamma_{i,u}\left(t\right)+\gamma_{u}}\right),$$
where *W* is the channel bandwidth pre-allocated for each PD before the data transmission mode is selected. For ease of expression, the signal-to-noise ratio (SNR) from the PD to the UAV is denoted by $\gamma_{i,u}\left(t\right)$, and $\gamma_{i,u}\left(t\right)=P_{i}\left(t\right)\cdot h_{i,u}\left(t\right)/\sigma^{2}$. The transmission power of the PD is denoted by $P_{i}\left(t\right)$, which selects different values in different time slots, and the parameter $\sigma^{2}$ is the variance of additive white Gaussian noise (AWGN). Similarly, the SNR of the data from the UAV to the LEO satellite is denoted by $\gamma_{u}$, and $\gamma_{u}=P_{u}\cdot h_{u}/\sigma^{2}$, where $P_{u}$ denotes the transmit power of the UAV.

When the PD chooses to forward data directly to the ground 5G base station, let the path loss between the PD *i* and the ground 5G base station as $l_{i,b}=l_{0}\cdot {\left(d_{i,b}\right)}^{-l}$, where $l_{0}$ denotes the channel gain per unit distance and *l* means the path loss index [31]. Furthermore, the corresponding channel gain is expressed as $h_{i,b}=10^{-\frac{l_{i,b}}{10}}$. Then, the data rate of the current time slot PD *i* to directly forward data to the ground 5G base station is
(5)$$R_{i,b}\left(t\right)=W\cdot \log_{2}\left(1+\frac{P_{i}\left(t\right)\cdot h_{i,b}}{\sigma^{2}}\right).$$

In summary, the actual data transmission rate of PD *i* at time slot *t* can be obtained as follows
(6)$$R_{i}\left(t\right)=x_{i,u}\left(t\right)\cdot R_{i,u}\left(t\right)+x_{i,b}\left(t\right)\cdot R_{i,b}\left(t\right).$$

The actual energy consumption of PD *i* at time slot *t* is given by
(7)$$E_{i}\left(t\right)=\left[x_{i,u}\left(t\right)\cdot P_{i}\left(t\right)+x_{i,b}\left(t\right)\cdot P_{i}\left(t\right)\right]\cdot \tau .$$

Therefore, the total long-term system energy efficiency is defined as
(8)$$U_{EE}=\frac{\sum_{t=1}^{T}\sum_{i=1}^{I}R_{i}\left(t\right)}{\sum_{t=1}^{T}\sum_{i=1}^{I}E_{i}\left(t\right)}.$$

### 3.2. Problem Modeling

Let $x_{u}=\left(x_{i,u}\left(t\right):\forall u\in U,i\in I,t\in T\right)$, $x_{b}=\left(x_{i,b}\left(t\right):\forall b\in B,i\in I,t\in T\right)$, and $P\left(t\right)=\left(P_{i}:\forall i\in I,t\in T\right)$, where U, B, I, and T denote the sets of UAVs, ground 5G base stations, PDs, and network operation time, respectively. The objective of this paper is to maximize the total long-term system energy efficiency and jointly optimize the PD association strategy ($x_{u}$ and $x_{b}$) and the power of PDs (P) within each time slot, so that the optimization problem can be expressed as
(9)$$\begin{array}{cc} P1: & \max_{x_{u},x_{b},P}\;U_{EE}\\ \;\; & s.t.\;C_{1}∼C_{7}. \end{array}$$
(10)$$\begin{array}{cc} \; & C_{1}:x_{i,u}\left(t\right)\in \left\{0,1\right\},\;\forall u\in U,i\in I,t\in T,\\ \;\; & C_{2}:x_{i,b}\left(t\right)\in \left\{0,1\right\},\;\forall b\in B,i\in I,t\in T,\\ \;\; & C_{3}:x_{i,u}\left(t\right)+x_{i,b}\left(t\right)\leq 1,\;\forall u\in U,b\in B,i\in I,t\in T,\\ \;\; & C_{4}:\sum_{i=1}^{I}x_{i,u}\left(t\right)\leq I_{U},\;\forall u\in U,i\in I,t\in T,\\ \;\; & C_{5}:\sum_{i=1}^{I}x_{i,b}\left(t\right)\leq I_{B},\;\forall b\in B,i\in I,t\in T,\\ \;\; & C_{6}:0\leq P_{i}\left(t\right)\leq P_{\max},\;\forall i\in I,t\in T,\\ \; & C_{7}:R_{i}\left(t\right)\geq R_{th},\;\forall i\in I,t\in T. \end{array}$$

$C_{1}$ and $C_{2}$ denote that the PD association strategy is a binary strategy. $C_{3}$ indicates that at most one data transmission way is selected for each PD at each time slot, i.e., either the data are forwarded directly to the ground 5G base station or the data are uploaded to the UAV and then forwarded to the LEO satellite. $C_{4}$ and $C_{5}$ represent that there is an upper limit on the number of PDs associated to the UAV or ground 5G base station per time slot, $I_{U}$ or $I_{B}$, respectively. $C_{6}$ shows that the transmission power of each PD is limited by the maximum power $P_{\max}$. $C_{7}$ means that the transmission rate of each PD needs to meet the minimum transmission rate requirement.

## 4. Problem Solution

Since the above optimization problem contains binary variables, it can be seen that it is a nonlinear constrained programming problem with a high solution complexity. Therefore, we decompose the joint optimization problem (9) into two subproblems by solving the device association strategy and the power control scheme within each time slot separately, so as to find the optimal solution of the objective function.

### 4.1. Device Association Strategy

In this subsection, we solve the device association subproblem in each time slot to obtain the device association strategy in each time slot; here, we do not consider the device power optimization, so the PDs are assigned equal power, and then, the optimization problem P1 in each time slot is transformed as follows
(11)$$\begin{array}{cc} \;\mathrm{SP}1: & \;\;\;\;\;\;\;\;\;\;\;\max_{x_{u}\left(t\right),x_{b}\left(t\right)}\;\frac{\sum_{i=1}^{I}R_{i}\left(t\right)}{\sum_{i=1}^{I}E_{i}\left(t\right)}\\ \;\;s.t.\; & C_{1}:x_{i,u}\left(t\right)\in \left\{0,1\right\},\;\forall u\in U,i\in I,t\in T,\\ \; & C_{2}:x_{i,b}\left(t\right)\in \left\{0,1\right\},\;\forall b\in B,i\in I,t\in T,\\ \; & C_{3}:x_{i,u}\left(t\right)+x_{i,b}\left(t\right)\leq 1,\;\forall u\in U,b\in B,i\in I,t\in T,\\ \; & C_{4}:\sum_{i=1}^{I}x_{i,u}\left(t\right)\leq I_{U},\;\forall u\in U,i\in I,t\in T,\\ \; & C_{5}:\sum_{i=1}^{I}x_{i,b}\left(t\right)\leq I_{B},\;\forall b\in B,i\in I,t\in T. \end{array}$$

The objective of subproblem SP1 is to solve the device association strategy. Since the association matrices $x_{u}\left(t\right)$ and $x_{b}\left(t\right)$ are subject to the constraints $C_{1}$–$C_{5}$, the solution dimension is too high if solving for the whole association matrix. To reduce the solution complexity, this section adopts the genetic algorithm [32] for solving SP1. The genetic algorithm is able to achieve a global stochastic search to obtain the global optimal solution by simulating the evolutionary phenomenon of species in the biological world. The algorithm works in an encoded manner, can search multiple peaks in parallel, and does not operate on the parameters themselves, which has good operability and strong robustness [33].

The genetic algorithm mainly consists of chromosomes, genes, population individuals, fitness function, a selection process, a replication process, a crossover process, and a mutation process. Among them, a chromosome represents a feasible solution to the problem to be solved, while all the constituent elements in each feasible solution can be seen as genes in the chromosome, and the carrier of the chromosome is called the population individual; the fitness function represents the degree of adaptation of the population individuals to the survival environment and can be used to measure the chromosomal merit of the population individuals in the evolutionary process; the selection process selects the better individuals from the current population so that they can continue to evolve without being eliminated; the replication process indicates that in each evolutionary process, the best individuals from the previous generation are retained and replicated intact to the next generation; the crossover process can be seen as a hybridization of genes in the chromosomes of population individuals, in which the selected parent chromosomes are randomly matched two by two to generate a new set of chromosomes according to the crossover method, and the crossover process is used to retain the good genes in each evolutionary process so that the result obtained is close to the local optimal solution; while the mutation process is used to achieve the global optimal solution by randomly selecting several genes on the chromosomes for random modification, thus introducing new genes into the current gene sequence in order to break the search limit to find the global optimal solution [34].

In problem SP1, by combining the device and UAV association strategy $x_{u}\left(t\right)$ and the device and ground 5G base station association strategy $x_{b}\left(t\right)$, we can represent the overall device association decision as $A\left(t\right)=x_{u}\left(t\right)\times x_{b}\left(t\right)$, and
(12)$$a_{i,j}\left(t\right)=\left\{x_{i,u}\left(t\right),x_{i,b}\left(t\right):\forall j\in U\cup B,i\in I,t\in T\right\}.$$

At this time, the matrix A(t), as a feasible solution in problem SP1, can be used as a chromosome in the genetic algorithm, and each element in A(t) is called a gene on the chromosome of the population individual. In addition, the fitness function is set to the total system energy efficiency in each time slot as follows
(13)$$fitness=\frac{\sum_{i=1}^{I}R_{i}\left(t\right)}{\sum_{i=1}^{I}E_{i}\left(t\right)}.$$

Based on this, in order to solve the problem SP1, we propose a genetic algorithm-based device association strategy, as summarized in Algorithm 1. First, the population is initialized (lines 2–3), and a certain number of feasible solutions to problem SP1 are randomly generated as for the initial population, and the genes of each individual chromosome are binary encoded, while each feasible solution needs to ensure that the constraints $C_{1}$–$C_{5}$ are satisfied. Next, the fitness function (lines 5–7) is calculated for the population individuals, and their fitness degree is calculated separately using the fitness function (12). Then, the population individual selection operation (line 8) is performed, and the elitist retention strategy is used to select the individuals with the largest fitness function values to be retained, called elite individuals, and then, the roulette strategy is used to determine the level of selection probability according to the fitness function values, where individuals with high fitness have a higher chance of being selected, while individuals with low fitness are more likely to be eliminated in the offspring population, so that a certain number of individuals can be selected to be retained, and the rest of the individuals with low fitness can be eliminated. In the replication process (line 9), elite individuals are replicated into the next generation population. Then, the crossover process (line 10) is carried out, setting a certain crossover probability $p_{Crossover}$. For the parent chromosomes to be crossed, a set of two is used, and each set is subjected to a single-point crossover operation: a random crossover point is set in the individual chromosome, thus dividing the chromosome into two parts, with the left and right sides of the offspring chromosome coming from the parent chromosome, respectively. For the newly generated offspring individuals, it is necessary to determine whether they satisfy the $C_{1}$–$C_{5}$ constraints, and if so, they are retained; otherwise, the new individuals need to be discarded. The next mutation is performed (line 11), setting a certain mutation probability of $p_{Mutation}$. Since each gene position is a binary variable, the mutation of gene positions is achieved by inverting the variables from 0 to 1. In addition, it should be noted that each device is associated with at most one location in the same time slot, i.e., a UAV or a ground 5G base station, and thus, the gene position which is already ’1’ needs to be ‘0’ subsequently before flipping for the mutation process. Similarly, the new individual after mutation also needs to be judged whether it meets the $C_{1}$–$C_{5}$ constraints. By the above operation, the next generation population can be obtained (line 12) and then return to line 5 for a loop until convergence or the maximum number of iterations is reached.
**Algorithm 1:** Genetic algorithm-based device association strategy1:**Input:** Device set I, UAV set U, ground 5G base station set B and channel state information.2:**Output:** Optimal device association strategy $\left\{x_{u}\left(t\right),x_{b}\left(t\right)\right\}$.3:**Initialize:** Population size $N_{number}$, crossover probability $p_{Crossover}$, mutation probability $p_{Mutation}$, maximum number of iterations $N_{iteration}$ and f=1.4:Random generate initial population $J_{f}$.5:**repeat**6:**for** each population individual in $J_{f}$ **do**7:   Calculate the fitness function using (12).8:**end for**9:Selective manipulation of individuals using elitist retention strategies and roulette wheel strategies.10:Replication of elite individuals into the next generation of populations.11:For all parental chromosomes that are selected, a set of two is used, and each set is crossed over according to the crossover probability $p_{Crossover}$ to generate new offspring to be added to the next generation population.12:According to $p_{Mutation}$, genes in the chromosomes of non-elite individuals of the parent are mutated, and the mutated individuals are inserted into the next generation population.13:Combining lines 8–11 yields the new generation of populations $J_{f+1}$.14:f=f+1.15:**until** $f>N_{iteration}$.16:**Return** $\left\{x_{u}\left(t\right),x_{b}\left(t\right)\right\}$.

By performing Algorithm 1 within each time slot, the optimal device association strategy can be obtained with equal power allocation for each device.

### 4.2. Power Control Scheme

In this subsection, based on the optimal device association strategy obtained in the previous section, the power of PDs is optimized to further improve the system energy efficiency. The optimization problem is transformed into SP2.
(14)$$\begin{array}{cc} \;\mathrm{SP}2: & \;\;\;\;\;\;\;\;\;\;\;\max_{P(t)}\;f\left(P(t)\right)=\frac{\sum_{i=1}^{I}R_{i}\left(t\right)}{\sum_{i=1}^{I}E_{i}\left(t\right)}\\ \;\;s.t.\; & C_{6}:0\leq P_{i}\left(t\right)\leq P_{\max},\;\forall i\in I,t\in T,\\ \; & C_{7}:R_{i}\left(t\right)\geq R_{th},\;\forall i\in I,t\in T. \end{array}$$

It can be seen that the objective function of subproblem SP2 is nonconvex and therefore cannot be solved using convex optimization theory. Furthermore, we use the idea of the simulated annealing algorithm and improve the algorithm to solve the power control scheme. The simulated annealing algorithm is derived from the solid annealing principle and is a probability-based algorithm [35]. The algorithm starts from a certain higher initial temperature, along with the decreasing temperature parameter, and combines the probabilistic sudden jump property to randomly find the global optimal solution of the objective function in the solution space; i.e., the local optimal solution can probabilistically jump out and eventually converge to the global optimum. The simulated annealing algorithm is an optimization algorithm that can effectively avoid falling into a serial structure of local minima and eventually converge to the global optimum by giving the search process a time-varying probabilistic jump property that eventually converges to zero [36].

The simulated annealing algorithm consists of two main parts, namely the annealing process and the Metropolis algorithm, which correspond to the outer and inner loops, respectively. The outer loop is the annealing process, which brings the solid to a high initial temperature $T_{0}$ and then decreases the temperature in a certain proportion according to the cooling factor, and when the termination temperature $T_{end}$ is reached, the cooling ends, i.e., the annealing process is finished. The Metropolis algorithm is an inner loop; i.e., at each temperature, it iterates $L_{iteration}$ times to find the optimal value of the energy at that temperature, i.e., the maximum value of the objective function for the subproblem SP2. Specifically, the Metropolis algorithm accepts new feasible solutions with probability instead of using a completely deterministic rule, also known as the Metropolis criterion: if the current temperature is $T_{now}$, where $T_{0}\leq T_{now}\leq T_{end}$, in the iterative process, the current optimal solution wants to change from *a* to *b* when the function values of the two feasible solutions are calculated as f(a) and f(b). If f(b)>f(a), the function value increases; then, this optimal solution transfer is accepted, i.e., the probability of the current optimal solution transfer to *b* is 1. If f(b)≤f(a), it means that the system deviates from the global optimal value position further. At this time, the Metropolis algorithm does not immediately discard it, but it carries out a probabilistic operation to generate a uniformly distributed random number *∂* in the interval [0, 1]. If $\partial <e^{\frac{E(b)-E(a)}{T_{now}}}$, then accept the new feasible solution *b* as the current optimal solution; otherwise, keep the original feasible solution *a* as the current optimal solution and go to the next step, and the loop repeats. By the above way, the Metropolis criterion can prevent falling into local optimal solutions [37].

However, when the network size, the number of iterations, and the difference between the initial and termination temperatures are large, the simulated annealing algorithm will cool down particularly slowly, and it is likely to waste too much time to find a better solution. Therefore, in solving the subproblem SP2, we improve the simulated annealing algorithm to solve the power control scheme to accelerate the cooling process of the system. Specifically, if the system finds a better feasible solution during the iterative search at the current temperature, we give the system an additional cooling rate $v_{decrease}$ to accelerate the system annealing; when the iteration at that temperature ends, if the system does not find a better feasible solution, then the system continues the annealing process with the original cooling rate $V_{decrease}$ to find a better solution.

We summarize the improved power control scheme based on the simulated annealing algorithm in Algorithm 2. In Algorithm 2, the system is first initialized and the initial solution is randomly generated, the value of the objective function corresponding to the initial solution is calculated (lines 2–4), and the process of outer loop annealing is started. Next, at the current temperature, it enters an inner loop to generate new feasible solutions and uses the Metropolis criterion to determine whether to accept the new feasible solutions until the end of the inner loop when the number of iterations is reached (lines 9–22). Then, the temperature is updated using the improved strategy (lines 23–26) and re-entered into the inner loop until the current temperature is lower than the termination temperature, the outer loop is ended and the algorithm is terminated to obtain the optimal power control scheme for all PDs under the current time slot.
**Algorithm 2:** Improved power control scheme based on simulated annealing algorithm1:**Input:** PD power maximum $P_{\max}$ and the minimum transmission rate requirement of the PDs $R_{th}$.2:**Output:** Optimal power control scheme {P(t)}.3:**Initialize:** Initial temperature $T_{0}$, original cooling rate $V_{decrease}$, additional cooling rate $v_{decrease}$, termination temperature $T_{end}$ and the number of iterations $L_{iteration}$.4:Randomly generate a random initial solution $p_{1}$ to the subproblem SP2.5:Calculate $f\left(p_{1}\right)$.6:**repeat**7:**for** temperature $T_{0}$ **do**8:   Set k=0, r=0.9:   **repeat**10:   Randomly generate a random initial solution $p_{2}$ to the subproblem SP2.11:   Calculate $f\left(p_{2}\right)$.12:   Calculate $\Delta f=f\left(p_{2}\right)-f\left(p_{1}\right)$.13:   **if** Δf>0 **then**14:     $p_{1}=p_{2}$.15:     r=r+1.16:   **else**17:     Generate a uniformly distributed random number *∂* in the interval [0,1].18:     **if** $e^{\frac{\Delta f}{T_{0}}}>\partial$ **then**19:        $p_{1}=p_{2}$.20:     **end if**21:   **end if**22:   k=k+1.23:   **until** $k>L_{iteration}$.24:   **if** r>0 **then**25:     $T_{0}=T_{0}\times v_{decrease}$.26:   **end if**27:   $T_{0}=T_{0}\times V_{decrease}$.28:**end for**29:**until** $T_{0}<T_{end}$.30:**Return** $p_{1}$.

### 4.3. Joint Device Association and Power Control Online Optimization Algorithm

Combining Algorithms 1 and 2, we propose a Joint Device Association and Power Control Online Optimization (JDAPCOO) algorithm of 5G and satellite converged networks for transmission and distribution scenarios, as shown in Algorithm 3. Within each time slot, the device power is first initialized, the device association strategy is solved according to Algorithm 1, and then the device power is solved using Algorithm 2 according to the obtained device association strategy. The algorithm terminates until both the device association strategy and the power control scheme are solved for each time slot during the network operation time. For simplicity, we add the Algorithm 3 flowchart in Figure 3.
**Algorithm 3:** Joint Device Association and Power Control Online Optimization (JDAPCOO) algorithm1:**Input:** Device set I, UAV set U, ground 5G base station set B, channel state information, PD power maximum $P_{\max}$ and the minimum transmission rate requirement of the PDs $R_{th}$.2:**Output:** Optimal device association strategy $\left\{x_{u},x_{b}\right\}$ and optimal power control scheme {P}.3:**repeat**4:**for** each time slot **do**5:   Initialize PDs’ power and decide the device association strategy $x_{u}\left(t\right)$ and $x_{b}\left(t\right)$ according to Algorithm 1.6:   Based on $x_{u}\left(t\right)$ and $x_{b}\left(t\right)$ obtained from Algorithm 1, the power control scheme P(t) is decided using Algorithm 2.7:   Update channel state information.8:   t=t+1.9:**end for**10:**until** t>T.11:**Return** $\{x_{u},x_{b}$ and P}.

## 5. Simulation Results

This section verifies the effectiveness of the proposed JDAPCOO algorithm through MATLAB simulations. The considered scenario has PDs randomly distributed in a 1km×1km geographical area, a UAV flies in a circle with a radius of 300 m at a fixed altitude with a fixed flight speed of 20 m/s, an LEO satellite and a ground 5G base station are used as cloud server data centers to collect the PDs’ data, as shown in Figure 4 as the projection schematic of the simulation scenario in the horizontal plane. Due to space constraints, the detailed system model simulation parameters are shown in Table 1.

Furthermore, the relevant parameters of the algorithm proposed in this paper are shown in Table 2 and Table 3, respectively.

Figure 5 and Figure 6 show the convergence of Algorithms 1 and 2 for different numbers of PDs with the 100th time slot as an example, respectively. From the figures, we can see that as the number of PDs increases, the number of iterations required to find the global optimal solution will also surge, and the convergence time of the algorithm will become longer, so we can set a suitable limited number of iterations or annealing speed to obtain a suboptimal solution close to the optimal one.

In order to illustrate the effectiveness of the JDAPCOO algorithm proposed in this paper, we compare the JDAPCOO algorithm proposed in this paper with the following three algorithms:Equal power allocation algorithm [38]: The power of all PDs is set as the same value within the maximum power range. For the convenience of comparison with the JDAPCOO algorithm in this paper, the device association strategy still adopts the algorithm proposed in this paper.Random power allocation algorithm [39]: The power of all PDs is determined randomly within the maximum power range. Like the equal power distribution algorithm [38], the device association strategy still adopts the algorithm proposed in this paper.Random device association algorithm: All PDs are randomly associated with the UAV or ground 5G base station. In order to show the superiority of the device association strategy in this paper, the power control scheme of PDs is the same as that in this paper.

Figure 7 shows the curve of the total system energy efficiency changing with the number of PDs. The total energy efficiency of the four algorithms all shows an increasing trend with the increasing number of PDs. However, the JDAPCOO algorithm proposed in this paper is still improved compared with the other three algorithms. As the number of PDs continues to increase, the joint optimization algorithm proposed in this paper simultaneously optimizes the device association strategy and power control, thus bringing significant performance improvement. Moreover, in order to more clearly demonstrate the superiority of the algorithm proposed in this paper compared with other algorithms, the energy efficiency values of different algorithms under different number of PDs are shown in Table 4. When the number of devices is 50, the total energy efficiency of the algorithm proposed in this paper is about 1, 5 and 8 times higher than that of the equal power allocation algorithm, random power allocation algorithm and random device association algorithm, respectively.

Figure 8 shows the relationship between total energy efficiency and the maximum power $P_{\max}$. It can be seen that for small $P_{\max}$, the total energy efficiency of the JDAPCOO algorithm proposed in this paper increases with the increase of $P_{\max}$, indicating that a higher transmission power threshold is needed to achieve the overall energy efficiency optimization. However, when $P_{\max}$ reaches a certain value, the total energy efficiency increases slowly and gradually converges, and the total energy efficiency of the other two algorithms is always lower than that of the proposed algorithm. Therefore, by comparing the results of the three algorithms, it can be concluded that the JDAPCOO algorithm proposed in this paper has better performance.

Figure 9 shows that the total system energy efficiency increases with the increase of channel bandwidth, because a higher channel bandwidth corresponds to a better transmission rate and higher energy efficiency. In addition, compared with the other three algorithms, the proposed JDAPCOO algorithm has better performance.

Figure 10 considers the impact of link characteristics on the total system energy efficiency. By changing the noise power of the link between PDs and UAV or ground 5G base station, the total system energy efficiency will change. In Figure 10, with the increase of noise power, the total system energy efficiency decreases, because the larger noise power will lead to the decline of transmission performance and then the total system energy efficiency. Compared with other algorithms, it can also be seen that the algorithm proposed in this chapter can provide higher total energy efficiency.

## 6. Conclusions

In this paper, we investigate the resource scheduling and energy efficiency optimization problem of 5G and satellite converged networks for transmission and distribution scenarios. A Joint Device Association and Power Control Online Optimization (JDAPCOO) algorithm is proposed with the aim of maximizing the total long-term system energy efficiency while ensuring the minimum transmission rate requirement of PDs. The proposed JDAPCOO algorithm can make an asymptotically optimal device association strategy and power control scheme based on the current network state information, and the simulation results confirm the superior performance of the JDAPCOO algorithm. Although the algorithm proposed in this paper has been improved in terms of energy efficiency, it has not considered the delay of data transmission of PDs. Therefore, future work will focus on the problem of delay and jitter guarantee for data transmission of PDs in 5G and satellite fusion networks so as to further meet the needs of power service carrying.

## Figures and Tables

**Figure 1 sensors-22-07085-f001:**
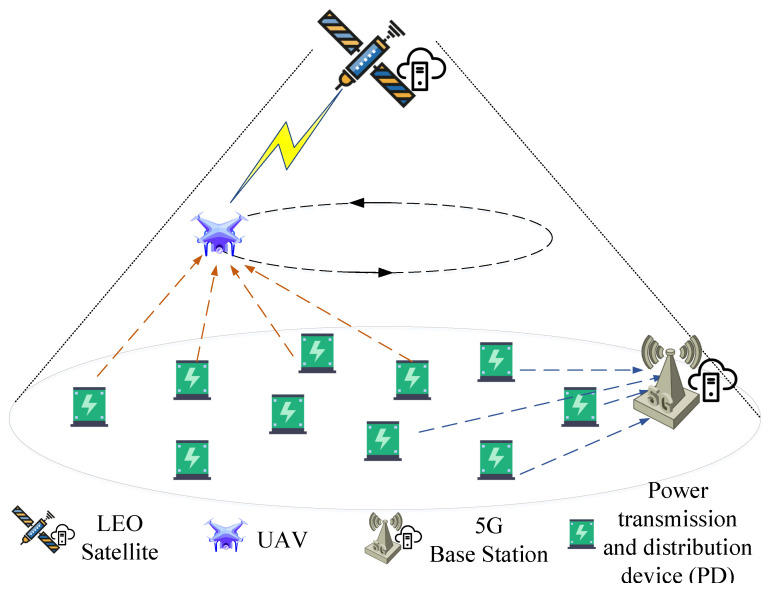
System Model.

**Figure 2 sensors-22-07085-f002:**
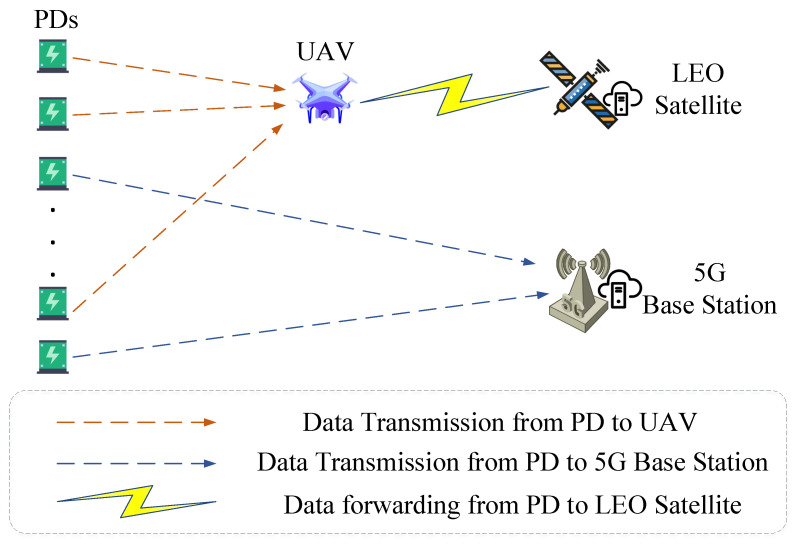
Communication mode of components participating in Figure 1.

**Figure 3 sensors-22-07085-f003:**
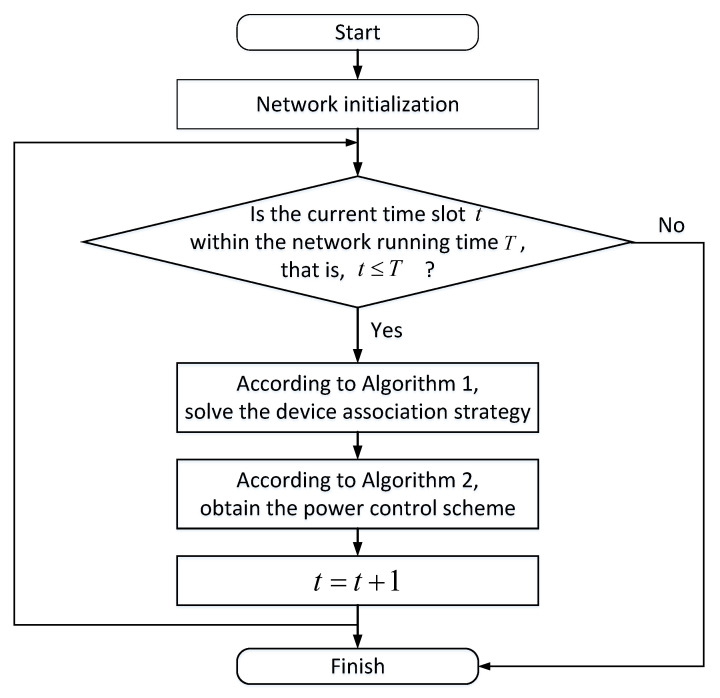
Algorithm 3 flowchart.

**Figure 4 sensors-22-07085-f004:**
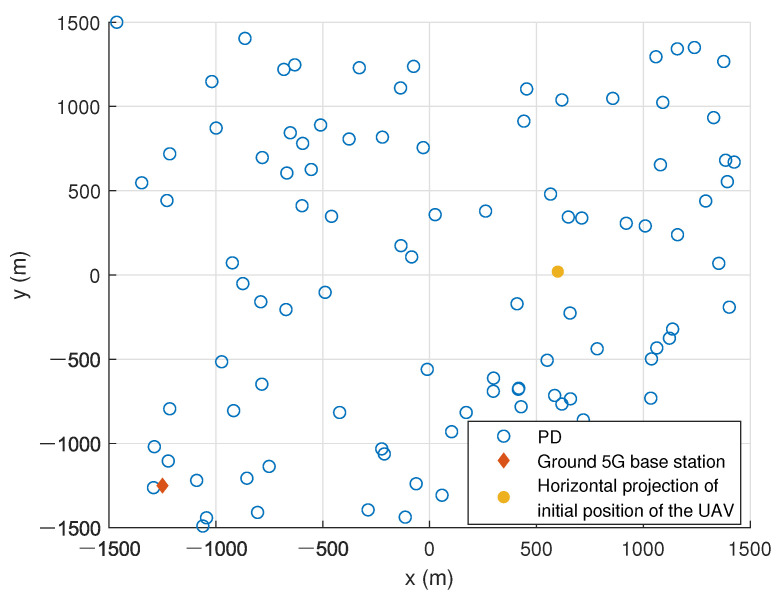
Projection schematic of the simulation scenario in the horizontal plane.

**Figure 5 sensors-22-07085-f005:**
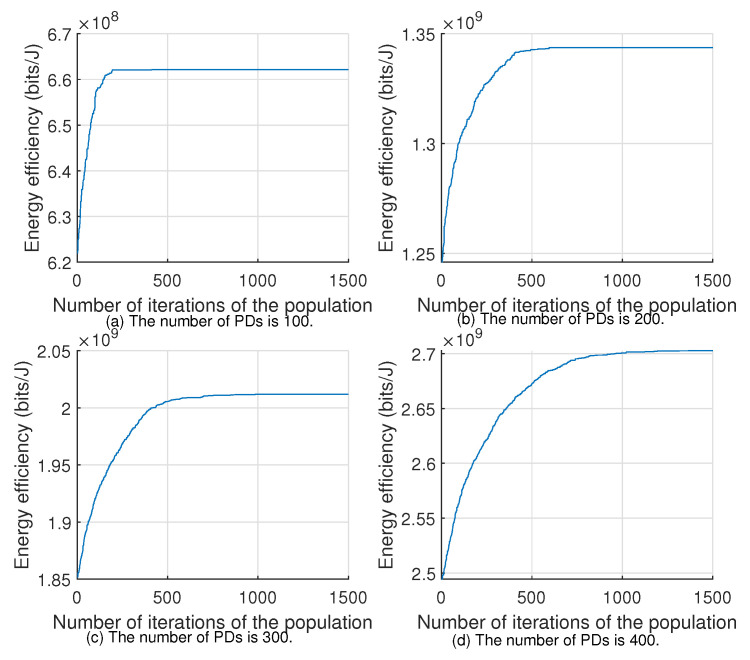
Convergence of Algorithm 1.

**Figure 6 sensors-22-07085-f006:**
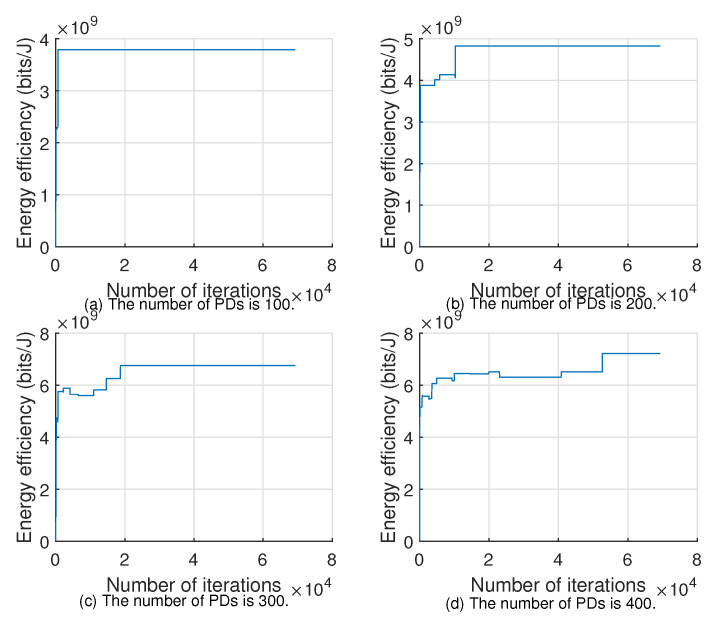
Convergence of Algorithm 2.

**Figure 7 sensors-22-07085-f007:**
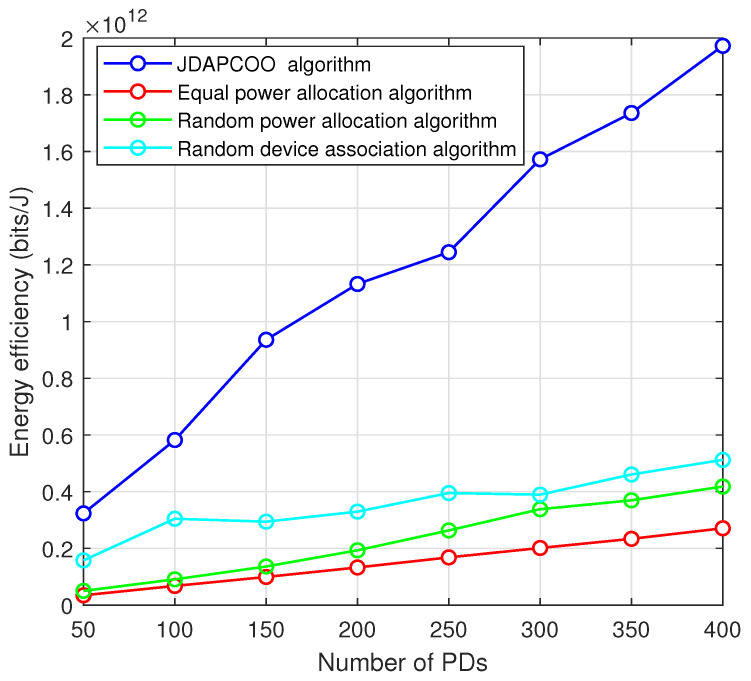
The relationship between the total system energy efficiency and the number of PDs.

**Figure 8 sensors-22-07085-f008:**
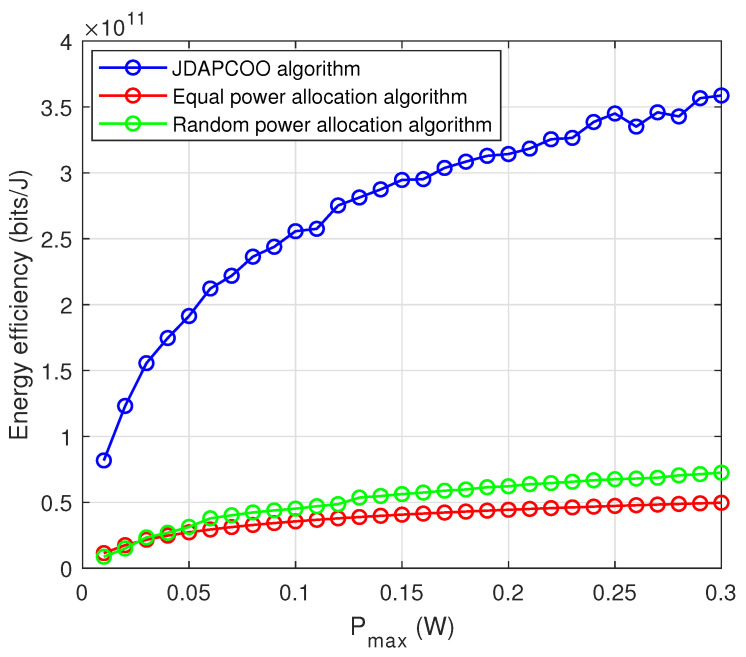
The relationship between the total system energy efficiency and the maximum power.

**Figure 9 sensors-22-07085-f009:**
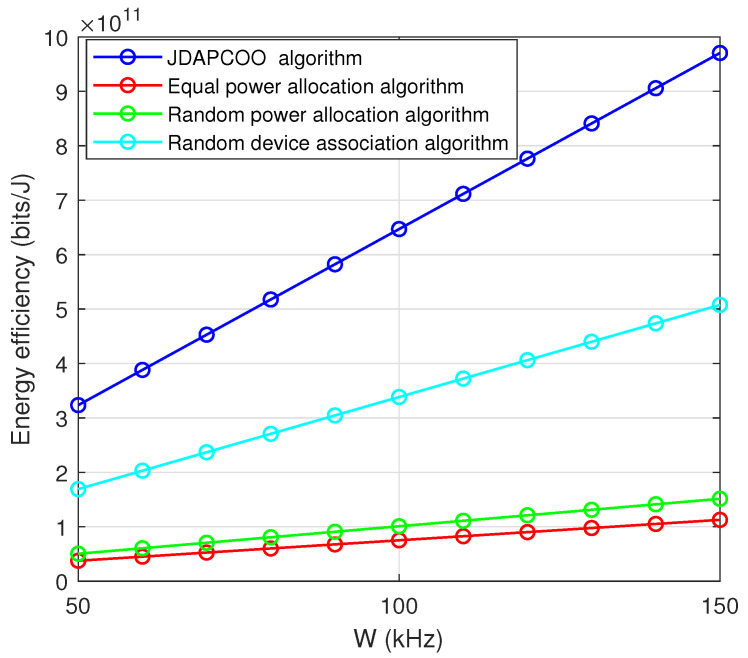
The relationship between the total system energy efficiency and the channel bandwidth.

**Figure 10 sensors-22-07085-f010:**
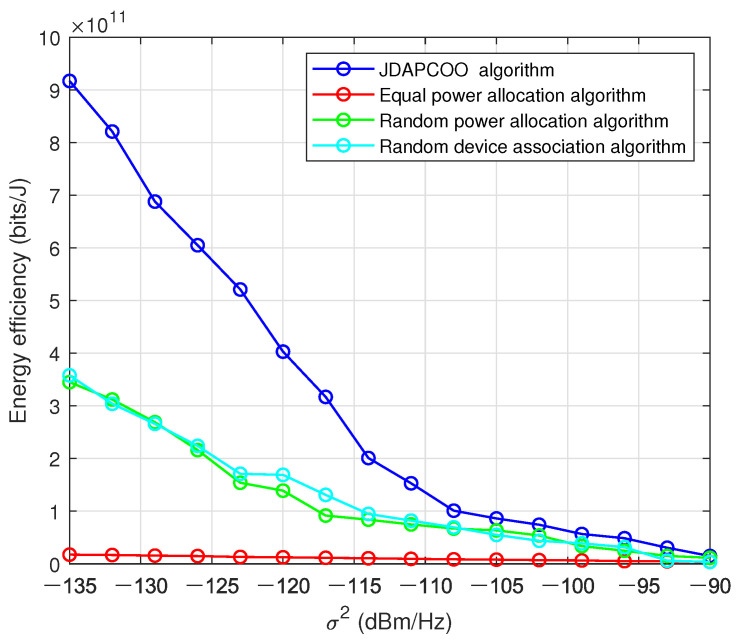
The relationship between the total system energy efficiency and the noise power.

**Table 1 sensors-22-07085-t001:** System model simulation parameters.

Parameter	Value	Parameter	Value
*I*	50∼400	*W*	90 kHz
*T*	100 s	$\sigma^{2}$	−127 dBm/Hz
τ	1 s	$P_{u}$	46 dBm
$H_{u}$	90 m	$I_{u}$	75
$h_{0}$	1.4 $\times \;10^{-4}$	$I_{B}$	75
$H_{g}$	200 km	$P_{\max}$	40 dBm
*l*	4	$R_{th}$	10 Mbps

**Table 2 sensors-22-07085-t002:** Algorithm 1 simulation parameters.

Parameter	Value
$N_{number}$	90
$p_{Crossover}$	0.1
$p_{Mutation}$	0.3
$N_{iteration}$	1500

**Table 3 sensors-22-07085-t003:** Algorithm 2 simulation parameters.

Parameter	Value
$T_{0}$	1
$V_{decrease}$	0.999
$v_{decrease}$	0.999
$T_{end}$	0.5
$L_{iteration}$	6932

**Table 4 sensors-22-07085-t004:** Total system energy efficiency of different algorithms with different PDs.

Different Algorithms	Number of PDs							
	50	100	150	200	250	300	350	400
JDAPCOO								
algorithm	3.23 $\times \;10^{11}$	5.82 $\times \;10^{11}$	9.36 $\times \;10^{11}$	1.13 $\times \;10^{12}$	1.25 $\times \;10^{12}$	1.57 $\times \;10^{12}$	1.74 $\times \;10^{12}$	1.97 $\times \;10^{12}$
Equal power								
allocation algorithm	3.46 $\times \;10^{10}$	6.76 $\times \;10^{10}$	9.93 $\times \;10^{10}$	1.33 $\times \;10^{11}$	1.68 $\times \;10^{11}$	2.01 $\times \;10^{11}$	2.34 $\times \;10^{11}$	2.71 $\times \;10^{11}$
Random power								
allocation algorithm	5.02 $\times \;10^{10}$	9.08 $\times \;10^{10}$	1.36 $\times \;10^{11}$	1.93 $\times \;10^{11}$	2.63 $\times \;10^{11}$	3.38 $\times \;10^{11}$	3.70 $\times \;10^{11}$	4.18 $\times \;10^{11}$
Random device								
association algorithm	1.58 $\times \;10^{11}$	3.05 $\times \;10^{11}$	2.94 $\times \;10^{11}$	3.30 $\times \;10^{11}$	3.95 $\times \;10^{11}$	3.90 $\times \;10^{11}$	4.61 $\times \;10^{11}$	5.12 $\times \;10^{11}$

## Data Availability

Not applicable.

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
