# Peer review of "JDAPCOO: Resource Scheduling and Energy Efficiency Optimization in 5G and Satellite Converged Networks for Power Transmission and Distribution Scenarios"

_sensors, 2022, doi:10.3390/s22187085_

Round 1

Reviewer 1 Report

The paper proposes Fifth Generation (5G) and satellite converged network architecture for power transmission and distribution scenarios, where power transmission and distribution devices (PDs) can choose to forward power data to cloud server data center via ground networks or space-based networks for power grid regulation and control. However, the following comments should be considered in this round. 

(1) The presentation of acronyms should be provided with the same consistency for all in the paper. Line 42, the first letters are uppercase, while the rest of the other acronyms are small case letters.

(2) In related works, authors are recommended to add the author’s name before citing when appearing first sentences.

 (3) Authors are recommended to separate the last paragraph as a new one. (Therefore, in order …………)

 (4) In 3.1. System Model, Authors are recommended to add items/points to explain the components participating in the system according to Fig.1.

 (5) In 4. Problem Solution, Authors are recommended to add a new figure (Fig.2) to show how to communicate components participating in the system according to Fig.1.

 (6) In 4. Problem Solution, Authors are recommended to add a flowchart for simplicity.

 (7) Authors are recommended to change the acronym JDP to JDAPCOO algorithm -as an example-. Should be included in the first letter for all words in ((propose a joint Device association and Power control online optimization (JDP) algorithm)).

 (8) The title of the manuscript should be included the name of the proposed (e.g., JDP/JDAPCOO).

(9) In the Abstract, the proposed (e.g., JDP/JDAPCOO) should be included as well.

(10) In Conclusion, the authors are recommended to add the limitation of this algorithm used as well as future work.

(11) The authors are recommended to provide an improvement table to show how your algorithm  outperforms others.

(12) In the abstract, the improvement percentage of this work, as a result, is missing.

Round 2

Reviewer 1 Report

Thanks to answer and revise the concerns according to the previous round. I have confirmed, that authors have to pay extra attention to following instructions. This, my decision will be accepted in present form.

Reviewer 2 Report

The authors have addressed my concerns, this version could be considered for publication.